# Derivation and validation of a model to predict treatment failure among under five children with severe community acquired pneumonia who are admitted at Debre Tabor specialized comprehensive hospital

**Muluken Chanie Agimas**[1]*, **Tigabu Kidie Tesfie**[1], **Nebiyu Mekonnen Derseh**[1], **Amare Kassaw**[2]

**1** Department of Epidemiology and Biostatistics, Institute of Public Health, College of Medicine and Health Science, University of Gondar, Gondar, Ethiopia, **2** Department of Pediatrics and Child Health Nursing, College of Health sciences, Debre Tabor University, Debre Tabor, Ethiopia

* mulukensrc12@gmail.com

## Abstract

### Background

Severe community-acquired pneumonia related treatment failure is persistence of features of severe pneumonia after initiation of antimicrobial therapy or a worsening clinical condition within 48–72 hours of the commencement of the antibiotics. Even though it is the most devastating public health problem in Ethiopia, there is no study to derivate and validate a model to predict treatment failure. To do this, nomogram was used to estimate the probability of treatment failure for each individual child and to classify their risk of treatment failure.

### Objective

to develop and validate the model to predict treatment failure among under five children with severe community-acquired pneumonia in Debre Tabor comprehensive specialized hospital.

### Method

A secondary analysis of the previously collected prospective follow-up study was used for further analysis among 590 under-5 children hospitalized with severe community-acquired pneumonia. The STATA version 17 software was used for analysis. Descriptive analysis was summarized by frequency and percentage. A multivariable binary logistic regression was also conducted, and the model performance was evaluated using the receiver operating characteristics curve with its area under the curve and calibration curve. Internal validation of the model was assessed using the bootstrap technique. The decision curve analysis was also used to evaluate the usefulness of the nomogram.

**Data availability statement:** All relevant data are within the manuscript and its Supporting Information files.

**Funding:** The author(s) received no specific funding for this work.

**Competing interests:** The authors have declared that no competing interests exist.

**Abbreviation and acronym:** AUC: Area Under the Curve; CI: Confidence Interval; HIV: Human Immune Deficiency; SCAP: Severe Community Acquired pneumonia.

## Results

The incidence of treatment failure among severe community-acquired pneumonia children was 28.1% (95% CI: 24.7%–30.8%). The previous history of severe community-acquired pneumonia, abnormal pulse rate, chest indrowing, anemia, HIV status, and plural effusion remained for the final model. The area under the curve for the original model and validated model was 0.7719 (95%CI: 0.729, 0.815) and 0.7714 (95% CI: 0.728–0.82), respectively. The decision curve analysis showed that the nomogram had a better net benefit across the threshold probability.

## Conclusion

The incidence of treatment failure among children with severe community-acquired pneumonia was high in Debre Tabor comprehensive hospital. The previous history of severe community-acquired pneumonia, abnormal pulse rate, chest indrowing, anemia, HIV status, and plural effusion were the significant factors to develop the predictive model. The model had good discriminatory performance and internally valid. Similarly, the model has a good calibration ability with an insignificant loss of accuracy from the original. The models can have the potential to improve treatment outcomes in the clinical settings. But needs external validation before use.

## Background

Severe community-acquired pneumonia related treatment failure is persistence of features of severe pneumonia after initiation of antimicrobial therapy or a worsening clinical condition within 48–72 hours of the commencement of the antibiotics [1,2]. Community-acquired pneumonia is a bacterial, viral, and fungal lower respiratory infection that is not acquired at health facilities and can be manifested by persistent dry cough, elevated body temperature (more than 38 °C), chills, fatigue, difficulty breathing, rigours, and pleurisy chest pain [3]. It is the most frequent reason for hospitalization in children, and it costs a large amount of health system resources [4]. Severe community-acquired pneumonia (SCAP) is a life-threatening disease (the severe form of community-acquired pneumonia) that requires treatment in the intensive care unit and results in a higher mortality rate [5]. The prevalence of severe community-acquired pneumonia (SCAP) ranges from 5–35% in all CAP cases [6], and its fatality accounts for about 20–50% [7]. The rate of treatment failure among CAP patients is 11% [8,9]. The CAP is the most common cause of emergency care; about 75% of the CAP patients need emergency treatment [10].

Treatment failure in SCAP is a common condition in which an inadequate response to the antibiotic treatment can lead to the worsening of the disease and cause death [2]. Once treatment fails, the pathogen is disseminated out of the lung and causes empyema, meningitis, and endocarditis, which finally becomes a systemic disease and ends in death [2,11]. The timely risk classification and prediction of treatment failure for SCAP patients is crucial to rescuing therapeutic strategies and reversing the unfavorable outcome associated with SCAP [12].

Treatment failure causes prolonged hospitalization, reduces the quality of care, and places an places an economic burden on the health care system [13–17]. To reverse these impacts, identification of those children with a high risk of treatment failure is very important to close follow-up on their treatment response and early identification.

About 80% of the treatment failure of SCAP is because of the systemic inflammatory response [12]. The previous risk factor studies also showed that the most common causes of treatment failure in SCAP were host factors, appropriate treatment regimen [18], age, renal and cardiac comorbidity [19], infancy, malnutrition, severity of anaemia, rickets, lack of immunization, and hypoxia at baseline [20]. Even though the treatment failure associated with SCAP is the most devastating public health problem in Ethiopia, there is no sufficient studies to derivate and validate the model to predict the treatment failure among children with SCAP. Therefore, the current study aimed to assess the derivation and validation of a model to predict treatment failure among children with SCAP who are admitted to the Debre Tabor specialized comprehensive hospital intensive care unit.

## Method

### Study design, setting and period

A retrospective cohort study was used among children with SCAP who were admitted to the Debre Tabor specialized hospital intensive care unit. Debre Tabor (north central Ethiopia) is the administrative town of the south Gondar zone. It is located 103 km from the capital city of the Amhara region, namely Bahir Dar city, and 666 km from Addis Ababa (the capital city of Ethiopia). The geographic positioning system coordinates of Debre Tabor town are located at latitude (11.8500) and longitude (38.0167), and the average temperature ranges from 50°F to 76°F. Debre Tabor has a total of three health centers, one specialized comprehensive hospital, and six health posts. The data were conducted from November 27, 2022, to December 31, 2022.

### Population (domain) and data sources

The source population was all under five children with SCAP admitted and treated at any time in the Debre Tabor comprehensive specialized hospital intensive care unit, and under five children with SCAP during the study period were included in the study. We used the data from the previous research [21].

### Outcome of interest

Treatment failure (Yes, No).

### Independent variables

**Socio-demographic characteristics.** Age of the child and sex of the child.

**Baseline disorder/status.** Pulmonary tuberculosis, meningitis, diarrhoea, HIV status, dehydration, pleural effusion, anaemia, hypoglycemia, measles, chest indrowing, previous history of SCAP, weight for age, status at admission, pertussis, and pulse rate (all these independent variables were determined at the initial presentation).

### Operational definition

**Treatment failure.** Was measured as persistence of features of severe pneumonia (cough and/or difficult breathing, with or without fever, fast breathing, or a lower chest wall in drawing where their chest moves in or retracts during inhalation) after initiation of antimicrobial therapy or a worsening clinical condition within 48–72 hours of the commencement of the antibiotics [1,2]. The outcome was measured or determined during their hospital stay.

The **SCAP** was defined as a CAP associated with the presence of one major or two or more minor criteria. The major criteria included a need for mechanical ventilation and septic shock.

The minor criteria included systolic BP ≤ 90 mmHg, bilateral pneumonia or multilobar pneumonia, and PaO2/FIO2 ≤ 250 [22].

**Comorbidity.** A disease condition at admission in addition to SCAP like heart disease, childhood asthma, retroviral infection, tuberculosis, acute gastroenteritis, pertussis, anaemia, meningitis, measles, bronchitis, heart disease, urinary tract infection, or any chronic and acute disease [23].

**Hypoglycemia.** According to the American Diabetic Association (ADA), hypoglycemia proposes a threshold of < 70 mg/dl or 3.9 mmol/L [24].

Child weight was measured by an electronic digital weight scale for children who were comfortable being measured alone, but for children who were unable to be measured alone, we used mother-child weight combined measurement, and then the mother's weight was measured to calculate the child's weight through subtraction. The weight measurement was recorded to the nearest 0.1 kg. A child weight-for-age Z-score was classified as z-scores < -3 and >=-3 of the WHO standard curve [25].

**Status of admission.** The consciousness level during admission as consciousness and unconsciousness [21].

## Sampling procedure and sampling technique

We used the data source, the sample size of 590 from the previous study conducted at Debre Tabor Comprehensive Hospital [21]. Thus, for more information about the sampling procedure and sampling technique, please refer to this published article [21].

## Data collection procedures and data quality assurance

Further analysis of the previously collected data was used. But during the primary data collection, the authors used interviews to administer questionnaires to collect the data from the participants) by trained nurses. The tool was adapted from west Bengal and India [23,26–29] and modified for the Ethiopian context. The treatment failure status of the children was obtained from the patient chart using the operational definition. To assure the validity of the tool, the English version of the questionnaire was translated to Amharic, which is the local language of the area, then back to English. A test was also conducted among 5% (30 participants) of the sample size outside of the study area to avoid information contamination. One-day intensive training was given for data collectors and supervisors to collect reliable and validated data. For the purpose of the current research, the data were accessed on July 20, 2023, from the primary authors of the previously published work [30] and the last author of the current study.

## Data processing and analysis

The data were entered into Epi-data version 4.6.02 and exported to STATA software version 17 for cleaning, recoding, missing data checking, and further analysis. There was no missing data in the collected data, and the descriptive analysis was summarized by frequency and percentage. Using a p-value of less than 0.25 as a cutoff point, simple (bi-variable) binary logistic regression analysis was conducted to select the candidate variables for multivariable analysis. After the candidate variables were selected, a multivariable binary logistic regression analysis (backward step-wise logistic regression) was conducted. Variables with a p-value < 0.05 were declared significant factors. After repeated runs of the model, a model with the calibration (the assumption of Hosmer and Lemishow test), the better discriminatory power of area under the curve (AUC) in the receiver operating characteristics curve (ROC curve), a model with a small number of variables that best fit (parsimony model selection), plausibility, and easily interpretability were the guiding rules to select and develop a prediction model for treatment failure

of SCAP. After the model was developed, its sensitivity, specificity, positive predictive value, negative predictive value, and accuracy were analyzed. The AUC was also used to assess the discriminatory power and performance of the prediction model. A nomogram tool was developed based on prognostic determinants found significant in the prediction model. The bootstrapping technique (1000 random samples with replacement) was analyzed to evaluate the internal validity of the developed prediction model. The sensitivity, specificity, positive predictive value, negative predictive value, beta coefficients, and AUC were compared before and after bootstrapping, and then the optimism coefficient was evaluated. The developed nomogram model can estimate the risk of treatment failure for individual children. A decision curve analysis was used to assess the clinical and public health significance of the nomogram [28]. Risk classification was also analyzed (as a low-risk versus high-risk risk of treatment failure) based on the maximum Youden index with its corresponding optimal cutoff point.

## Ethical declaration

**Ethical approval and consent to participate.** Previously, during primary data collection, ethical approval was obtained from the Debre Tabor University ethical review board with the protocol number Dtu/RP/186/15, and written informed consent was obtained from each mother or legal guardian. But for the current research, we used the previously collected data. The owner of the data was Mr. Amare Kassaw, who was the primary author of the previously published work [21] and the last author of the current study.

## Results

### Baseline characteristics

About 580 participants, with a response rate of 98.3%, participated in the study. About 312 (46.2%) of them were male. Ninety-one (15.69%) and 74 (12.79%) participants were impaired (unconscious) and infected with HIV, respectively. Regarding the history of previous illness, 141 (24.31%) of under-five children have had a history of previous SCAP. Furthermore, 51 (8.79) of them were already infected with tuberculosis and known at the baseline (Table 1).

### Incidence of treatment failure and prognostic predictors of SCAP

The overall incidence of SCAP treatment failure among children who were admitted to Debre Tabor comprehensive specialized hospital was 28.1% (95% CI: 24.7%, 30.8%). After conducting the bi-variable analysis one by one, a total of 15 predictors were the candidate variables for further multivariable analysis (Table 2).

### Predictive model for SCAP treatment failure

Using the 17 predictors like pulmonary tuberculosis, meningitis, diarrhea, HIV status, dehydration, plural effusion, anaemia, hypoglycemia, measles, chest indrowing, weight for age, admission status, pertussis, pulse rate, sex of child, age of child, and previous SCAP, the multivariable logistic regression was analyzed, and finally, predictors like anaemia, previous history of SCAP, plural effusion, pulse rate, HIV status, measles, and chest indrowing were remained and used for the reduced model (Table 3). Finally, the nomogram was developed using the reduced model. The AUC of the original model was 0.7719 (95%CI: 0.729, 0.815) (Fig 1), and the calibration/model fitness at an angle of $45^0$ showed that there was no difference between the predicted probability and the observed probability with a p-value of 0.917 (Fig 2). To estimate the performance of the original model, the optimal cut point was determined by the maximum Youden index value. Thus, the Youden index value was 0.4389 (Supporting File 1). At these

**Table 1. Baseline characteristics of under-five children with SCAP admitted at Debre Tabor specialized comprehensive hospital, northwest, Ethiopia, 2022.**

| Variable | Category | Frequency | % |
|---|---|---|---|
| Sex of child | Male | 312 | 46.2 |
| | Female | 268 | 53.8 |
| Age of child | <24 months | 287 | 50.5 |
| Tuberculosis | >=24 months | 293 | 49.5 |
| | Yes | 51 | 8.79 |
| | No | 529 | 91.21 |
| Meningitis | Yes | 74 | 12.76 |
| | No | 506 | 87.24 |
| Diarrhea | Yes | 145 | 25 |
| | No | 435 | 75 |
| Infected with HIV | Yes | 74 | 12.76 |
| | No | 336 | 57.93 |
| | Unknown | 170 | 29.31 |
| Plural effusion | Yes | 74 | 12.76 |
| | No | 506 | 87.24 |
| Anemia | Yes | 97 | 16.72 |
| | No | 483 | 83.28 |
| Hypoglycemia | Yes | 115 | 19.83 |
| | No | 465 | 80.17 |
| Measles | Yes | 77 | 13.28 |
| | No | 503 | 86.72 |
| Chest indrowing | Yes | 232 | 40 |
| | No | 348 | 60 |
| Previous history of SCAP | Yes | 141 | 24.31 |
| | No | 439 | 75.69 |
| Weight for age Z-score value | ≥-3 | 407 | 70.17 |
| | <-3 | 173 | 29.83 |
| Status at admission | Impaired | 91 | 15.69 |
| | Conscious | 489 | 84.31 |
| Pertussis | Yes | 94 | 16.21 |
| | No | 486 | 83.79 |
| Pulse rate | Normal | 338 | 58.28 |
| | Abnormal | 242 | 41.72 |
| Dehydration | Yes | 154 | 26.55 |
| | No | 426 | 73.45 |

cutoff points, the model had an accuracy of 76.38% (95% CI: 73%–80%), a sensitivity of 55.83% (95% CI: 48%–64%), a specificity of 84.41% (95% CI: 81%–88%), a positive predictive value (PPV) of 58.33% (95% CI: 59%–66%), and a negative predictive value of 83.02% (95% CI: 79%–86%). To evaluate the internal validity of the model, to avoid over interpreting, and to reduce the optimistic result from the original model, bootstrapping (1000 bootstrap samples with replacement) was used, and thus the AUC for the internally validated model was 0.7714 (0.728, 0.82) with an optimism coefficient of 0.0005 (Fig 1). The calibration of the internally validated model showed that there was agreement between the predicted and observed probability with a p-value of 0.549 (Fig 3). The very small optimism coefficient of a well-calibrated model can be

**Table 2. Bivariable logistic regression analysis to develop and validate a prediction model for SCAP treatment failure among under five children in Debre Tabor comprehensive specialized hospital, 2022.**

| Variable | β coefficient | p-value | β (95%CI) | |
|---|---|---|---|---|
| | | | LCI | UCI |
| PTB | | | | |
| Yes | 0.9139 | 0.002 | 0.33094 | 1.498 |
| No | | | | |
| Meningitis | | | | |
| Yes | 0.4465 | 0.088 | -0.0659 | 0.959 |
| No | | | | |
| Diarrhea | | | | |
| Yes | 0.405 | 0.049 | 0.0013 | 0.809 |
| No | | | | |
| Infected with HIV | | | | |
| Yes | 1.423 | <0.001 | 0.83 | 2.017 |
| No | 0.463 | 0.043 | 0.014 | 0.912 |
| Unknown | | | | |
| Dehydration | | | | |
| Yes | 0.328 | 0.107 | -0.071 | 0.727 |
| No | | | | |
| Plural effusion | | | | |
| Yes | 1.496 | <0.001 | 0.99 | 2 |
| No | | | | |
| Anemia | | | | |
| Yes | 0.932 | <0.001 | 0.482 | 1.382 |
| No | | | | |
| Hypoglycemia | | | | |
| Yes | 0.492 | 0.026 | 0.059 | 0.924 |
| No | | | | |
| Measles | | | | |
| Yes | 1.018 | <0.001 | 0.528 | 1.508 |
| No | | | | |
| Chest indrowing | | | | |
| Yes | 1.122 | <0.001 | 0.748 | 1.496 |
| No | | | | |
| Previous history of SCAP | | | | |
| Yes | 1.597 | <0.001 | 1.19 | 2.004 |
| No | | | | |
| WAZ | | | | |
| Z score ≥ -3 | -0.603 | 0.002 | -0.985 | -0.219 |
| Z-score < -3 | | | | |
| Status at admission | | | | |
| Impaired | 1.068 | <0.001 | 0.608 | 1.529 |
| Conscious | | | | |
| Pertussis | | | | |
| Yes | 0.836 | <0.001 | 0.38 | 1.293 |
| No | | | | |
| Pulse rate | | | | |
| Abnormal | 0.313 | 0.093 | -0.052 | 0.678 |

*(Continued)*

**Table 2.** (Continued)

| Variable | β coefficient | p-value | β (95%CI) | |
| --- | --- | --- | --- | --- |
| | | | LCI | UCI |
| Normal | | | | |
| Sex | | | | |
| Male | 0.16 | 0.67 | -0.45 | 0.87 |
| Age | 0.51 | 0.42 | -0.72 | 0.98 |

**Table 3. Multivariable binary logistic analysis for the prediction of SCAP treatment failure SCAP among under five children at Debre Tabor specialized comprehensive hospital northwest, Ethiopia, 2022.**

| Predictors | Multivariable analysis | | |
| --- | --- | --- | --- |
| | Original β (95%CI) | P-value | Bootstrap β (95%CI) |
| Pulmonary tuberculosis(yes) | NA | | |
| Meningitis (yes) | NA | | |
| Diarrhea (yes) | NA | | |
| Infected with HIV (yes) | 1.14 (0.465, 1.81) | 0.001 | 1.14 (0.64,1.64) |
| Pulse rate (abnormal) | 0.604 (0.181, 1.027) | 0.005 | 0.603(0.183, 1.023) |
| Dehydration (yes) | NA | | |
| Plural effusion (yes) | 0.795 (0.216, 1.374) | 0.007 | 0.765 (0.715, 0.815) |
| History of SCAP (yes) | 1.08 (0.636,1.531) | <0.001 | 1.07 (0.42, 1.72) |
| Anemia (yes) | 0.546 (0.027, 1.065) | 0.039 | 0.55 (0.43, 0.67) |
| Hypoglycemia (yes) | NA | | |
| Chest indrowing (yes) | 0.917 (0.486, 1.349) | <0.001 | 0.915 (0.495, 1.335) |
| Status during admission (impaired) | NA | | |
| Pertussis (yes) | NA | | |
| Weight for age z-score (≥-3) | NA | | |
| Measles | 0.645 (0.089, 1.2) | 0.023 | 0.647 (0.092, 1.202) |

**Fig 1. Area under the ROC curve for SCAP treatment failure.**

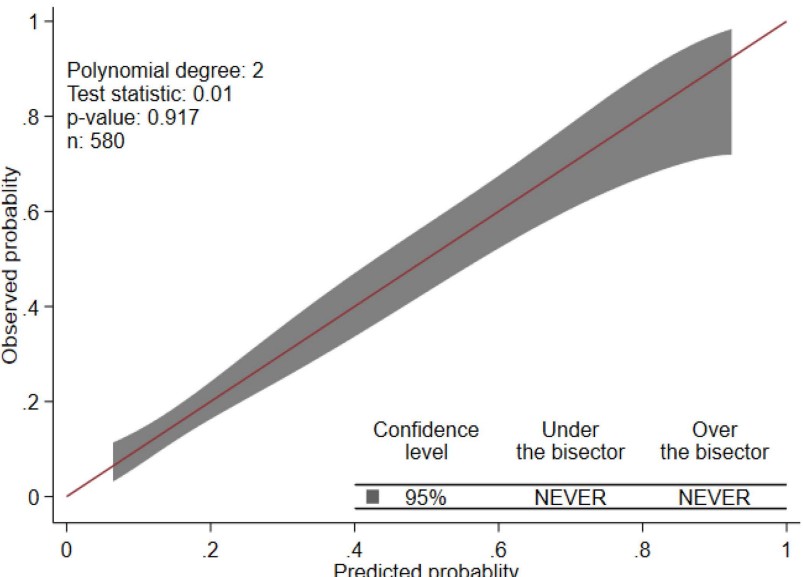

**Fig 2. Predicted versus observed probability of SCAP treatment failure (calibration plot for original model).**

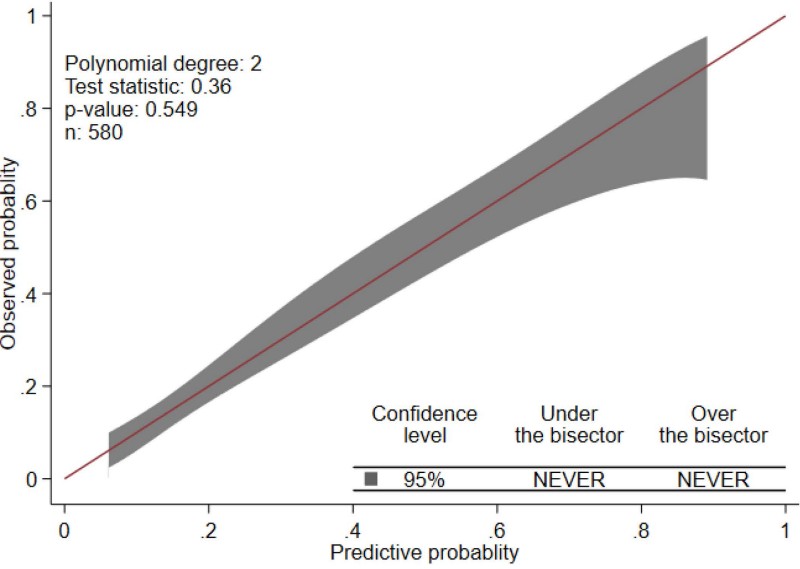

**Fig 3. Predicted versus observed probability of SCAP treatment failure (calibration plot for validated model).**
$P$ (SCAP treatment failure) $= 1/e^{-(-2.67 + 1.14*\text{HIV status yes} + 0.765*\text{plural effusion (yes)} + 0.55*\text{anemia (yes)} + 0.647*\text{measles (yes)} + 0.915*\text{chest indrowing (yes)} + 1.07 *\text{history of SCAP (yes)} + 0.603*\text{pulse rate}}$ (abnormal) (Table 3).

applied to this model for the new sample. The risk prediction of SCAP treatment failure for the internally validated model using linear predictors.can be stated as follows:

## Nomogram for prediction of SCAP treatment failure

The nomogram was developed using a reduced model using variables with better predictive performance, biological plausibility, and easy interpretation. To develop the nomogram, seven

predictors, such as anaemia, the previous history of SCAP, pulse rate, HIV status, measles, plural effusion, and chest indrowing, were included. The role of each predictor was also presented in the form of a nomogram division score. Using the total score of the SCAP treatment failure in the nomogram, the probability of treatment failure for each individual child can be calculated (Fig 4).

## Treatment failure risk classification using nomogram

To classify the risk of treatment failure, the maximum Youden index (max J = 0.4389) was used. At this value, the corresponding cut-off point was 0.3585. Using this cutoff point, the risk was classified as low risk for treatment failure (<0.3585) and high risk for treatment failure (≥0.3585). Thus, 126 (48.3%) of the patients were in the high-risk treatment failure group (Table 4).

## Decision curve analysis

The standardized net benefit analysis was evaluated using decision curve analysis, and thus, across the probability, the nomogram had the better net benefit (Fig 5).

## Discussion

In the current study, an attempt has been made to derivate and validate the model to predict the treatment failure associated with SCAP among under-five children who were admitted to

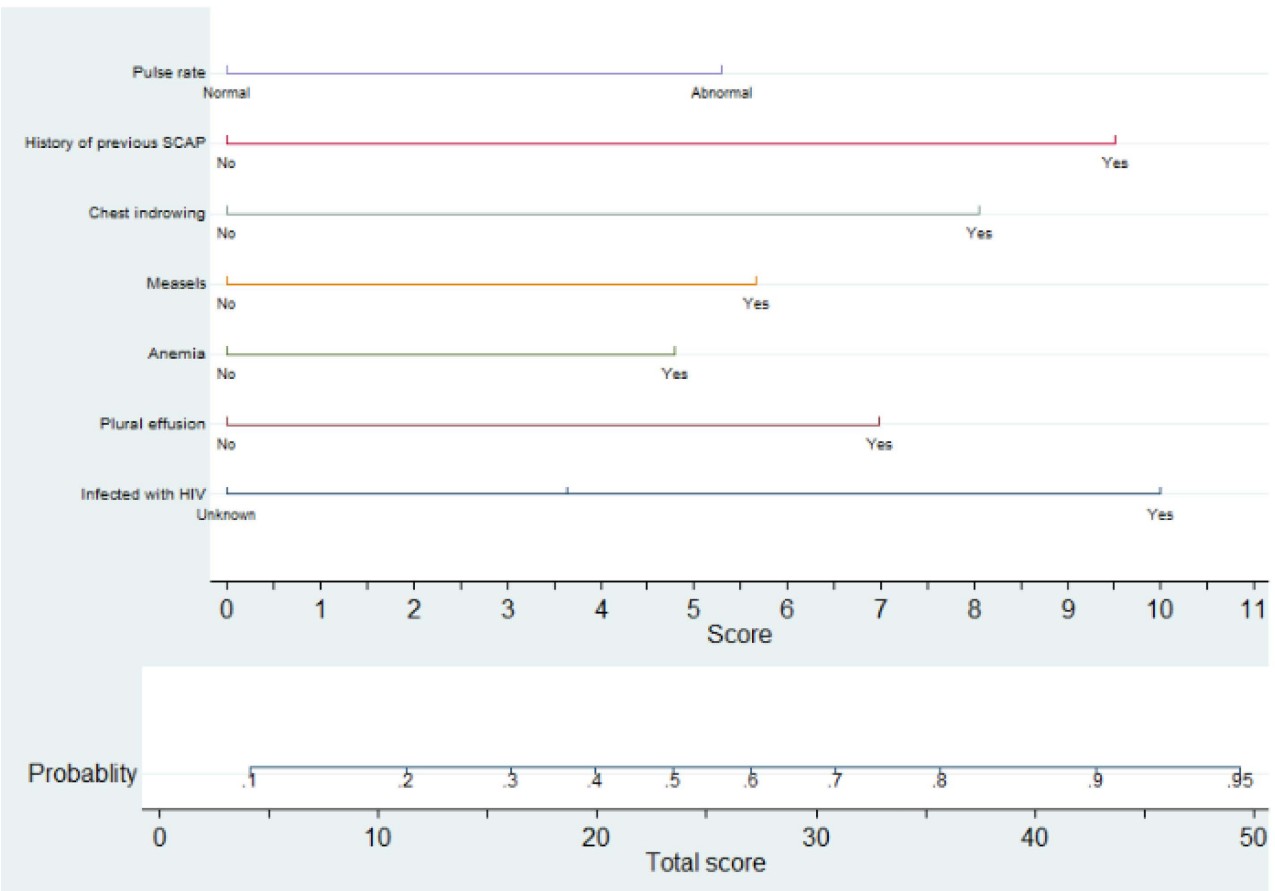

**Fig 4. Nomogram developed for the prediction of SCAP treatment failure among under five children.**

**Table 4. Risk classification of treatment failure for SCAP among under five children admitted at Debre Tabor comprehensive specialized hospital northwest, Ethiopia, 2022.**

| Risk category | Frequency | Incidence of treatment failure |
|---|---|---|
| Low risk (<0.3485) | 319 | 37(11.6%) |
| High risk (≥0.3585) | 261 | 126 (48.3%) |

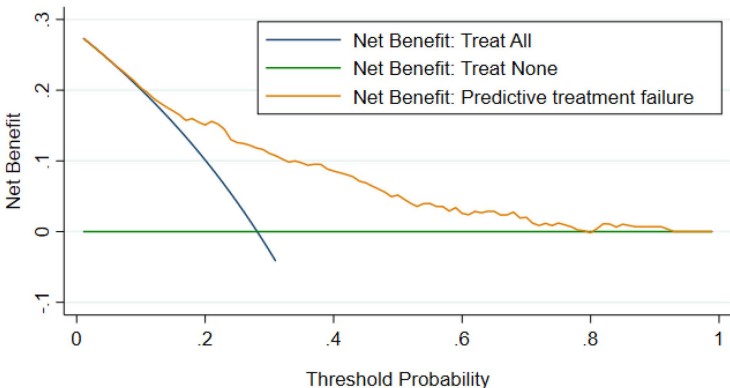

**Fig 5. Decision curve plot showing standardized net benefit of the nomogram against threshold probability.**

Debre Tabor Comprehensive Specialized Hospital. This paper is studying the same domain in the same setting for the prediction of mid-term outcomes. The overarching objective would still be the same for selecting patients that require more intensive monitoring. To achieve this objective, a total of 580 under-5 children who were admitted to the Debre Tabor specialized comprehensive hospital during the follow-up period were considered for analysis. Thus, the incidence of SCAP treatment failure among under-five children was 28.1% (95% CI: 24.7%, 30.8%). The finding was higher than a study conducted in India (18.2%) [20]. This might be because of the difference in socio-economic status, health care access, and health-seeking behaviour between the two countries. The maximum Youden index value with its corresponding cut-off point was the guiding rule to classify the risk of treatment failure for SCAP, and thus 48.3% of patients were at high risk of SCAP treatment failure, which means at most 48.3% of the treatment failures may be identified earlier when using the prediction tool, followed by more intense monitoring of the high-risk group. At a value of the Youden index (max J = 0.4389) with its corresponding cut-off point of < 0.3585, it was classified as low risk for treatment failure (not performing closing monitoring), or about 11.6% of children with SCAP had a lower risk of treatment failure for not performing closing monitoring in pediatrics SCAP patients in the intensive care unit. The risk prediction model for SCAP treatment failure was developed using seven predictors: anaemia, previous history of SCAP, plural effusion, pulse rate, HIV status, measles, and chest indrowing. The discriminatory power of the model was evaluated using AUC, and its value was 0.7719 (95% CI: 0.729, 0.815) (Fig 1). The calibration or model fitness at an angle of 45⁰ showed that there was no difference between the predicted probability and the observed probability with a p-value of 0.917 (Fig 2). This well-calibrated model can avoid the risk of overestimating or underestimating the outcome of interest. To validate the model, to avoid over interpreting, and to reduce the optimistic result from the original model, the bootstrapping technique (1000 bootstrap samples with replacement) was used, and the adjusted AUC for the validated model was 0.7714 (0.728, 0.82) with an optimism coefficient of 0.0005

(Fig 1**).** The calibration of the validated model also showed that there was agreement between the predicted and observed probability with a p-value of 0.549 (Fig 3)**,** and the β coefficients of the bootstrapped model were nearly similar to those of the original model. Which implies that with the limited optimism coefficient (0.0005), our model can be well performed in a new sample. As far as our searching is concerned, there has been no research conducted on the derivation and validation of a risk prediction model for SCAP treatment failure among under-five children in Ethiopia, and thus it is hard to compare the findings with a model developed in the Ethiopian setting. Even though the model was slightly different in terms of the included predictors and biomarkers like white blood cells, lactate, procalcitonin, cortisol, B-type natri-uretic peptide, and C-reactive protein, the performance of our model was in line with a study conducted in a tertiary care university hospital in Portugal between December 2009 and March 2012 (AUC = 0.81, 95% CI: 0.74-0.91) [31]. Using biomarkers for risk prediction of SCAP treatment failure is very important for clinical decisions about treatment failure associated with SCAP, but in less developed countries like Ethiopia, the applicability and measurability of these biomarkers are very difficult in routine health care. Therefore, in the current study, the model considers the most applicable predictors in the Ethiopian context.

Our model has lower sensitivity (79%), higher specificity (71%), an inline positive predictive value (60%), and a negative predictive value (86%) than the previously developed model at the tertiary care university hospital in Portugal [31]. Developing a treatment failure SCAP prediction tool and classifying patients based on their risk of treatment failure is crucial to clinical decision-making by reducing the treatment failure rate among under-five children, but there is no tool to achieve this objective. Hereafter, it is essential for the evaluation and early identification of patients for their risk of SCAP treatment failure. To do this, our model was developed using predictors like previous history (SCAP), abnormal pulse rate, HIV status, anaemia, plural effusion, measles, and chest indrowing. These predictors are easily measurable and practicable for clinical decision-making in low-infrastructure countries like Ethiopia. Furthermore, our model has a better net benefit across the threshold probabilities. Which means our model has clinical and public health importance after external validation. Using easily measurable predictors in the routine health care system that suits Ethiopia was the strength of the study, but our model needs external validation before being used for clinical decisions. Assessment of clinical failure by the treating physician could be prone to information bias. Thus, it would be important for a future validation study to assess treatment failure inde-pendently, preferably by someone blinded to clinical characteristics at baseline. Additionally, the lack of related articles makes it more difficult to compare in detail with the other findings.

## Conclusion

The incidence of treatment failure among children with severe community-acquired pneu-monia was high in Debre Tabor comprehensive hospital. The previous history of severe community-acquired pneumonia, abnormal pulse rate, chest indrowing, anemia, HIV status, and plural effusion were the significant factors to develop the predictive model. The model had good discriminatory performance and internally valid. Similarly, the model has a good calibration ability with an insignificant loss of accuracy from the original. The models can have the potential to improve treatment outcomes in the clinical settings. But needs external validation before use

## Supporting information

**Supportive File 1.  Cut of point classifier.**
(XLSX)

**STROBE Statement. Checklist of items that should be included in reports of observational studies.**
(DOCX)

## Acknowledgments

The authors would like to give thanks to Debre Tabor University and Debre Tabor specialized comprehensive hospital.

## Author contributions

**Conceptualization:** Muluken Chanie Agimas, Tigabu Kidie Tesfie, Amare Kassaw.

**Data curation:** Muluken Chanie Agimas, Nebiyu Mekonnen Derseh.

**Formal analysis:** Muluken Chanie Agimas, Tigabu Kidie Tesfie.

**Investigation:** Muluken Chanie Agimas.

**Methodology:** Muluken Chanie Agimas.

**Software:** Muluken Chanie Agimas, Nebiyu Mekonnen Derseh, Amare Kassaw.

**Supervision:** Muluken Chanie Agimas, Amare Kassaw.

**Validation:** Muluken Chanie Agimas.

**Visualization:** Muluken Chanie Agimas, Nebiyu Mekonnen Derseh.

**Writing – original draft:** Muluken Chanie Agimas, Nebiyu Mekonnen Derseh.

**Writing – review & editing:** Muluken Chanie Agimas.

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
