## [Decision Letter · Decision Letter 0]

29 Jul 2024

PONE-D-23-36484Derivation and validation of a model to predict treatment failure among under five children with severe community acquired pneumonia who are admitted at Debre Tabor specialized comprehensive hospitalPLOS ONE

Dear Dr. Agimas,

Thank you for submitting your manuscript to PLOS ONE. After careful consideration, we feel that it has merit but does not fully meet PLOS ONE’s publication criteria as it currently stands. Therefore, we invite you to submit a revised version of the manuscript that addresses the points raised during the review process.

Please address the reviewers' comments below. ==============================

We look forward to receiving your revised manuscript.

Kind regards,

Mohammed Feyisso Shaka, MPH

Academic Editor

PLOS ONE

4. Please include your tables as part of your main manuscript and remove the individual files. Please note that supplementary tables should be uploaded as separate "supporting information" files.

Reviewers' comments:

Reviewer's Responses to Questions

**Comments to the Author**

1. Is the manuscript technically sound, and do the data support the conclusions?

Reviewer #1: Yes

Reviewer #2: Yes

Reviewer #3: Yes

2. Has the statistical analysis been performed appropriately and rigorously? 

Reviewer #1: Yes

Reviewer #2: Yes

Reviewer #3: No

3. Have the authors made all data underlying the findings in their manuscript fully available?

Reviewer #1: Yes

Reviewer #2: Yes

Reviewer #3: Yes

4. Is the manuscript presented in an intelligible fashion and written in standard English?

Reviewer #1: Yes

Reviewer #2: Yes

Reviewer #3: No

5. Review Comments to the Author

Reviewer #1: This paper is well-done, here is my comments

In the method section

Outcome of interest

Treatment failure (Yes, No)

Independent variables

Pulmonary tuberculosis, meningitis, diarrhea, HIV status, dehydration, plural infusion, anemia, hypoglycemia, measles, chest indrowing, previous history of SCAP, weight for age, status at admission, pertussis and pulse rate.

Comment

The independent variable better rearrange in theme like socio-demographic, health facility related factors, Individual related factors, …backward step-wise logistic regression analysis also follow this theme.

Reviewer #2: Comments to authors

Generally, this is an extensive, novel, and well written research. With the following very minor comments the research work is publishable.

Abstract:

 Check the grammar

 Write the objective of the research separately

 Write the figures in to two digits

Introduction:

In this section, the authors should address the following points

 What is the clinical importance of the research?

 What is the gap of the previous research?

Methods:

 Use the appropriate punctuation for line 123.

 In line 159 please specify the figure.

 What is the importance of accuracy by chance value?

 Why you used a systematic random sampling?

Results:

 Please include the table within the main manuscript

 The model “P (SCAP treatment failure) = 1/e-(-2.67 +1.14 *HIV status yes +0.765* plural effusion (yes) +0.55 193 *anemia (yes) +0.647*measles (yes) +0.915*chest indrowing (yes) +1.07 *history of SCAP (yes) + 194 0.603*pulse rate (abnormal) (Table-3)” is not well written please revise it.

 Why you report decision curve analysis?

Discussion:

 If similar studies are available so far, please include additional comparisons.

 In line 228, what does it mean 450?

 What is the strength of the study?

 How to compare the AUC with corresponding previous AUC findings?

 What is your recommendations for researcher in the future?

Reviewer #3: The authors need to have a nomogram model to show how much is loss of accuracy from the original one. The authors need to improve the discussion and conclusion part. You may see comments on the attached file or detailed below.

Abstract:

One of the most important aims of prognostic study is to find out prognostic determinants of the outcome of interest, however, your conclusion of your study in the abstract lacks this important facets of a conclusion.

Introduction:

The authors should have exclusively searched whether there was a study conducted so far on SCAP in the Ethiopia. The authors deny there is no single study done on the topic, however, there was one study here I dropped the link: https://pubmed.ncbi.nlm.nih.gov/36791115/. The study predicted poor outcomes of SCAP; treatment failure (prognostic outcome in the current study) was one of the composite outcomes in that study. Therefore, the authors in the current study should state and should properly justify that the current study is unique from the previous study; however, this is a flaw

Methods:

What by mean: “A secondary analysis of the previous prospective follow-up study was used among children with 92 SCAP who were admitted to the Debre Tabor specialized hospital intensive care unit” ? is this prospective study? What makes prospective if you use previously collected data from the hospital. You need to correct this.

Results:

Your original model resulted an accuracy of AUC 77.14%, however, the AUC of the validated model is 77.19%. How this logically acceptable. You do validation for the sake of minimizing over optimistic results. Validation technique is the same as adjustment the same as avoiding crude result for generalization. Logically, original results are over optimistic compared to validated ones, however, yours is controversial. This requires careful correction in case it is editorial problem or else.

The purpose of conducting prediction studies is to develop an easy to use risk stratification tool at the end of the study. In your case a nomogram tool; this is very important to have it. However, developing risk stratification tool can result in the loss of accuracy because it is simplification technique. This means always there will be a loss of accuracy from the original models. Therefore, researchers expected to develop a nomogram model, which means need to show the accuracy of nomogram model, so that you will be able to compare the results with original ones.

To construct the nomogram, it is good to have pulse rate and hemoglobin/hematocrit level as they are without classifying as normal vs. abnormal for the purpose of proper use of nomogram tool in the future. Therefore, include these variables in the model as continuous keeping the natural occurrences of the variables for correct interpretation of the tool. In addition, the author should put examples how to interpret the tool for the audience.

Discussion and Conclusion:

The authors didn’t well discuss their results with study findings and evidences in the country (in Ethiopia) and global context. The conclusion is a single sentence only about accuracy issue of the model. This very bad not make appropriate conclusion about the main

findings of the study based on the objectives of the study. Therefore, careful attention should be given and a significant change should be made to improve aforementioned parts of the manuscript.

6. PLOS authors have the option to publish the peer review history of their article (what does this mean? ). If published, this will include your full peer review and any attached files.

**Do you want your identity to be public for this peer review?** For information about this choice, including consent withdrawal, please see our Privacy Policy .

Reviewer #1: **Yes: ** Esubalew Tesfahun

Reviewer #2: No

Reviewer #3: **Yes: ** Zelalem Alamrew Anteneh

---

## [Decision Letter · Decision Letter 1]

3 Dec 2024

PONE-D-23-36484R1Derivation and validation of a model to predict treatment failure among under five children with severe community acquired pneumonia who are admitted at Debre Tabor specialized comprehensive hospitalPLOS ONE

Dear Dr. Agimas,

Thank you for submitting your manuscript to PLOS ONE. After careful consideration, we feel that it has merit but does not fully meet PLOS ONE’s publication criteria as it currently stands. Therefore, we invite you to submit a revised version of the manuscript that addresses the points raised during the review process.

We look forward to receiving your revised manuscript.

Kind regards,

Wen-Jun Tu

Academic Editor

PLOS ONE

Journal Requirements:

Reviewers' comments:

Reviewer's Responses to Questions

**Comments to the Author**

1. If the authors have adequately addressed your comments raised in a previous round of review and you feel that this manuscript is now acceptable for publication, you may indicate that here to bypass the “Comments to the Author” section, enter your conflict of interest statement in the “Confidential to Editor” section, and submit your "Accept" recommendation.

Reviewer #1: All comments have been addressed

Reviewer #3: (No Response)

2. Is the manuscript technically sound, and do the data support the conclusions?

Reviewer #1: Yes

Reviewer #3: Yes

3. Has the statistical analysis been performed appropriately and rigorously? 

Reviewer #1: Yes

Reviewer #3: Yes

4. Have the authors made all data underlying the findings in their manuscript fully available?

Reviewer #1: Yes

Reviewer #3: No

5. Is the manuscript presented in an intelligible fashion and written in standard English?

Reviewer #1: Yes

Reviewer #3: Yes

6. Review Comments to the Author

Reviewer #1: All my comments and suggesstion properly addressed, so that no addtional comments in this version.

Reviewer #3: Still, there are important issues to be addressed before acceptance for publication. I will attach my comments for each section of the manuscript to be revised.

7. PLOS authors have the option to publish the peer review history of their article (what does this mean? ). If published, this will include your full peer review and any attached files.

**Do you want your identity to be public for this peer review?** For information about this choice, including consent withdrawal, please see our Privacy Policy .

Reviewer #1: **Yes: ** Esubalew Tesfahun

Reviewer #3: **Yes: ** Zelalem Alamrew Anteneh

---

## [Decision Letter · Decision Letter 2]

19 Feb 2025

Derivation and validation of a model to predict treatment failure among under five children with severe community acquired pneumonia who are admitted at Debre Tabor specialized comprehensive hospital

PONE-D-23-36484R2

Dear Dr. Agimas,

We’re pleased to inform you that your manuscript has been judged scientifically suitable for publication and will be formally accepted for publication once it meets all outstanding technical requirements.

Kind regards,

Wen-Jun Tu

Academic Editor

PLOS ONE

Additional Editor Comments (optional):

Reviewers' comments:

Reviewer's Responses to Questions

**Comments to the Author**

1. If the authors have adequately addressed your comments raised in a previous round of review and you feel that this manuscript is now acceptable for publication, you may indicate that here to bypass the “Comments to the Author” section, enter your conflict of interest statement in the “Confidential to Editor” section, and submit your "Accept" recommendation.

Reviewer #1: All comments have been addressed

Reviewer #3: All comments have been addressed

2. Is the manuscript technically sound, and do the data support the conclusions?

Reviewer #1: Yes

Reviewer #3: Yes

3. Has the statistical analysis been performed appropriately and rigorously? 

Reviewer #1: Yes

Reviewer #3: Yes

4. Have the authors made all data underlying the findings in their manuscript fully available?

Reviewer #1: Yes

Reviewer #3: Yes

5. Is the manuscript presented in an intelligible fashion and written in standard English?

Reviewer #1: (No Response)

Reviewer #3: Yes

6. Review Comments to the Author

Reviewer #1: The authors addresed all my comments and concerns. No addtional comments and suggestion in this version.

Reviewer #3: (No Response)

7. PLOS authors have the option to publish the peer review history of their article (what does this mean? ). If published, this will include your full peer review and any attached files.

**Do you want your identity to be public for this peer review?** For information about this choice, including consent withdrawal, please see our Privacy Policy .

Reviewer #1: **Yes: ** Esubalew Tesfahun

Reviewer #3: **Yes: ** Zelalem Alamrew Anteneh

---

## [Editor Report · Acceptance letter]

PONE-D-23-36484R2

PLOS ONE

Dear Dr. Agimas,

I'm pleased to inform you that your manuscript has been deemed suitable for publication in PLOS ONE. Congratulations! Your manuscript is now being handed over to our production team.

Kind regards,

on behalf of

Dr. Wen-Jun Tu

Academic Editor

PLOS ONE